# Higher Prevalence of Food Insecurity and Psychological Distress among International University Students during the COVID-19 Pandemic: An Australian Perspective

**DOI:** 10.3390/ijerph192114101

**Published:** 2022-10-28

**Authors:** Seema Mihrshahi, Putu Novi Arfirsta Dharmayani, Janaki Amin, Alexandra Bhatti, Josephine Y. Chau, Rimante Ronto, Diana Turnip, Melanie Taylor

**Affiliations:** 1Department of Health Sciences, Faculty of Medicine Health and Human Sciences, Macquarie University, Macquarie Park, NSW 2109, Australia; 2School of Psychological Sciences, Faculty of Medicine Health and Human Sciences, Macquarie University, Macquarie Park, NSW 2109, Australia

**Keywords:** COVID-19, food insecurity, psychological distress, university students, international students

## Abstract

The COVID-19 pandemic and related disruptions have not only affected university students’ learning and academic outcomes, but also other issues, such as food security status, mental health and employment. In Australia, international students faced additional pressures due to sudden border closures and lack of eligibility for government-provided financial support. This study explored the experiences of domestic and international university students residing in Australia during the early stages of the COVID-19 pandemic across a range of outcomes. A cross-sectional online survey was conducted between July and September 2020 at Macquarie University in Sydney, Australia. The online survey included food insecurity status, mental health (psychological distress), disruptions to study, employment and sleep. A total of 105 students (*n* = 66 domestic and *n* = 39 international) completed the survey. Respondents reported having food insecurity (41.9%) and psychological distress (52.2%, with high and very high levels), with international students reporting significantly higher food insecurity (OR = 9.86 (95% CI 3.9–24.8), *p* < 0.001) and psychological distress scores (*t*(90) = 2.68, 95% CI: 1.30 to 8.81, *p* = 0.009) than domestic students. About one quarter of all respondents reported disruptions to study and employment status around the time of the survey. When asked what government support should be provided for international students, ‘financial aid’ was the most frequently suggested form of support. This research may help governments and educational institutions design appropriate support, particularly financial and psychological, for both international and domestic university students.

## 1. Introduction

The SARS-CoV-2 virus (COVID-19) was first reported as a cluster of cases of pneumonia of unknown origin in Wuhan, Hubei Province, China on 31 December 2019 [1]. On 11 March 2020, the World Health Organization (WHO) declared that COVID-19 was a global pandemic and posed an unprecedented health challenge [2,3]. To stop transmission of the virus, countries began enacting diverse public health and infrastructure measures. In Australia, these measures included strict lockdowns with suspension of face-to-face classes at schools and universities, disruptions to public transport routes, social gatherings, art and sporting events, and access to essential services only [4]. Similar to other countries, Australia also introduced border closures [4] and enforced quarantine and isolation for suspected cases and close contacts [5]. As a result, there have been drastic changes to normal patterns of life, including vast social and economic disruption.

One of the main challenges of this global disruption was the localised closure of educational institutions [6,7,8]. According to monitoring data from the United Nations Educational, Scientific and Cultural Organization (UNESCO), more than 87% of the global student population was severely impacted due to these closures [9]. Learning and teaching delivery was shifted from face-to-face to online modes. There is evidence that these changes resulted in detrimental effects on learning, health and wellbeing. Some studies report inadequate achievement of learning outcomes for some students due to the shift from face-to-face to online modes [10,11,12]. Other studies found higher levels of social isolation [11,13,14], poorer connections with peers [13,15] and concerns about employability [11,16].

Many students were also affected by job losses and as a result experienced financial hardship [17,18], struggling to cover costs of living, including rent and food [19,20,21]. These circumstances may have increased the likelihood of university students experiencing food insecurity. Several studies in the United States (US) have found higher rates of food insecurity among university students as a result of the pandemic [19,20,21].

Frustrations and anxiety caused by sudden changes to learning, financial and social conditions may have also led to higher rates of psychological distress and exacerbations of mental health problems [22]. There is emerging literature about the psychological impacts of the COVID-19 pandemic for students, and studies have been carried out in Jordan [23], Pakistan [24] and the US [11]. A common finding from these studies was that the transition to online learning caused an increase in psychological distress among university students due to lack of academic and peer connection, as well as a reduction in motivation and general dissatisfaction with online classes.

There is emerging evidence that, for international university students in many countries, the sudden closure of some international borders not only severely disrupted their learning and social activities but may have also impacted their sense of safety and belonging, leading to higher rates of psychological distress and making them particularly vulnerable to housing and food insecurity [25,26]. A qualitative study on Vietnamese students in the USA, for example, found that high levels of uncertainty and the impact of lockdown orders were likely to have substantial consequences in terms of psychological distress for these students [27].

In the Australian context, there is emerging evidence about the flow-on effects of disruptions on the mental health of university students [10,28,29,30]. In regard to food security, there is also recent evidence that the COVID-19 pandemic has impacted domestic and international students [31,32]. A few studies elsewhere show the disproportionately greater impact on the international student cohort [19,20]. In this study, we aimed to explore the experiences of domestic and international university students during a period of COVID-19 lockdown and restrictions in 2020 in Sydney, Australia. Our focus was psychological distress, food insecurity and their possible association. Our rationale was to provide governments, university governing bodies and other higher education institutions with valuable insights on what type of resources and services should be in place within Australia to support university students if similar events occur in the future.

## 2. Materials and Methods

### 2.1. Design and Participants

A convenience sample of currently enrolled university students aged ≥18 years from Macquarie University in Sydney, Australia, who resided in the Greater Sydney area, were recruited to participate in an online cross-sectional survey. Invitation emails were sent to course participants in the Master of Public Health, Medicine and undergraduate Public Health programs. Advertisements and notices were also posted online in student societies and groups across the university. Students who responded that they were interested in participating were given information about the study with a link to the participant information statement and online questionnaire. Students were asked to read the participant information statement and consent was obtained via a tick box on the online questionnaire before proceeding. All students were learning online at the time the survey was conducted due to lockdowns in the Sydney area.

A total of 142 students attempted the questionnaire; 10 students answered only one or two questions, and so were excluded. Twenty-seven students were also excluded due to one of the following reasons: failed to respond to questions about their current student status (domestic or international) (*n* = 22), or not living in Australia (*n* = 5). Data for the remaining 105 (73.9%) students are reported for rest of the analysis. Student ages ranged from 18 to 59 years, with 84.6% being female (88/105).

### 2.2. Data Collection

All data were collected via an online questionnaire (see Appendix A) using Qualtrics (Provo, UT, USA) and hosted on Macquarie University password-protected servers. The online survey was carried out between 16 July 2020 and 20 September 2020, approximately 4 months after COVID restrictions were introduced and students moved to online learning. Participants were asked to report their demographic characteristics, their current student status, and how the pandemic had affected their studies over the last semester in terms of time management and effort needed to keep up. We also included questions on food insecurity and mental health status. According to the Food and Agriculture Organization of the United Nations, food insecurity refers to conditions where a person experiences a “lack of regular access to enough safe and nutritious food for normal growth and development and an active and healthy life” [33]. Food insecurity was measured using the six-item United States Department of Agriculture Food Security Survey Module (USDA-FSSM) [34]. The Kessler psychological distress scale (K10) [35] was used to assess psychological distress, which was validated for use in Australia [36]. Short-answer questions about mental health conditions, employment, sleep quality, support payments and support for international students were also included.

### 2.3. Data Analysis and Coding

#### 2.3.1. Demographic and Impacts of COVID-19

The self-reported questionnaire consisted of demographic details such as age, sex, citizenship status (i.e., Australian, permanent residence, visa), spoken languages at home, study information (e.g., degree and faculty, study year), student status (i.e., international or domestic student), employment situation. It also included open-ended and Likert scale questionnaires asking about the impacts of COVID-19 on study, mental health conditions, and sleep.

#### 2.3.2. Food Security

For the USDA-FSSM food security questions [34], households were categorised as either: food secure (score 0–1), low food security (score 2–4) or very low food security (score 5–6), and data dichotomised to ‘food secure’ and ‘food insecure’ for the purpose of analyses (with ‘food insecurity’ status assigned to households with a score of two or higher).

#### 2.3.3. Psychological Distress

Psychological distress results (K10) were coded using the same method from the 2014-15 Australian National Health Survey [37], where participants were grouped into the following four levels of psychological distress:Low (scores of 10–15, indicating little or no psychological distress);Moderate (scores of 16–21);High (scores of 22–29);Very high (scores of 30–50).

### 2.4. Statistical Analysis

Demographic variables were presented as frequencies (n) and percentages (%) where categorical, and mean and standard deviation (SD) when the variables were continuous. To examine differences between international and domestic students with respect to food insecurity and psychological distress, we used Pearson’s Chi-Square test for categorical variables and the independent samples T-test for continuous variables. Two-sided *p* values of less than 0.05 were considered statistically significant. All statistical analysis was conducted by using IBM SPSS Statistics for Windows version 27.0 (Armonk, NY, USA: IBM Corp).

### 2.5. Ethical Approval

Ethical approval for this study was given by Macquarie University Human Research Ethics Committee (approval number 52020658117410).

## 3. Results

### 3.1. Demographic Characteristics

Table 1 shows the main demographic characteristics of the population. The mean age was 24.95 ± 6.61 years. Student ages ranged from 18 to 59, with most students (86.3%) aged under 30 years. The majority of students who completed the survey were female (84.6%). More than half of the students (61.6%) were Australian citizens or permanent residents. A total of 61.5% of students spoke English as their main language at home. In total, 57.1% were postgraduate students (including PhD candidates), and 41.9% undergraduate students. Most students were enrolled in the Faculty of Medicine, Health and Human Sciences (76.2%), of whom 51.3% were enrolled in the Master of Public Health course. Of the 105 students whose data was analysed according to student status, 66 students were domestic (62.9%) and 39 students were international (37.1%). Most students (91.4%) were studying full time. Of the international students, 87.2% were enrolled in postgraduate study. Just over half of the domestic students (59.1%) were enrolled in undergraduate study. The mean age of international students (26.63 ± 4.28 years) was higher than domestic students (23.95 ± 7.52 years) (*p* = 0.047).

### 3.2. Impact of COVID-19 on Studies

Table 2 shows how students perceived the pandemic affected their studies in the second half of 2020 (second semester, July to November). Most students (75%) felt they needed to invest more time or effort in their coursework than usual and reported feeling distracted. Only 58% felt they had done well in their studies. The majority of students (71%) felt they had the resources, such as technology and support, for their studies, and about two-thirds of students had a clear understanding of what they needed to do, and a similar proportion reported difficulty keeping up.

There were no statistically significant differences between international and domestic students with managing their studies to be completed on time, time investment for studying, feeling they did nothing instead of completing coursework, having a clear understanding of what expected to do, and having difficulty keeping up. However, there was a significantly higher proportion of domestic students reporting that they felt they did very well in their studies compared with international students. Similarly, a significantly higher proportion of domestic students reported they had the resources, such as technology and support, to do their coursework in comparison to international students.

### 3.3. Food Insecurity

Table 3 indicates that 27.6% of students responded that often or sometimes the food they bought did not last and they could not afford to buy more, and 34.3% stated they often or sometimes could not afford to buy balanced meals. Approximately one-fifth of students skipped meals and ate less than they should be eating (22.9%) because of lack of money for food. A total of 14.3% of students reported being hungry and not eating because there was not enough money for food. There was a generally low level of missing data for this hunger-based question, with 2.9–4.8% of students selecting “prefer not to answer”.

Table 4 shows food security levels according to the scoring criteria. Food insecurity among survey participants was relatively high, with 7.6% of students having very low food security and 34.3% having low food security, meaning 41.9% were categorised as ‘food insecure’. There was a significant association between student status (international vs. domestic) and food insecurity status (*Χ*^2^(1) = 26.844, *p* < 0.001), with the odds ratio for food insecurity in international students compared with domestic students being OR = 9.86 (95% CI 3.9–24.8), *p* < 0.001.

### 3.4. Psychological Distress

A total of 92 students (87.6%) provided a complete set of responses to the K10 psychological distress scale (Table 5). The mean score was 23.73 (SD = 8.78; 95% CI 21.91–25.55), which would be classified overall as a ‘high’ level of psychological distress. Table 5 compares the levels of psychological distress for domestic students, international students and all students. A large proportion of students were found to report psychological distress (52.2%) at a high or very high level in the past 30 days. More than a quarter of students (27.2%) reported ‘very high’ levels of psychological distress.

An independent samples *t*-test was performed to determine the mean difference for the K10 scores between two groups. The results indicated a statistically significant difference between domestic (M = 22.08, SD = 7.85) and international students (M = 27.13, SD = 9.71), *t*(90) = 2.68, *p* = 0.009. The average K10 score for international students was 5.05 points higher than its average for domestic students.

### 3.5. Food Insecurity and Psychological Distress

Food insecurity and psychological distress outcomes were also significantly associated. When compared according to their food security status, food-insecure students had a significantly higher prevalence of high psychological distress than food-secure students (60.4% vs. 39.6%), with the odds ratio for psychological distress in food-insecure students compared with food-secure students being OR = 8.07 (95% CI 3.0–21.8), *p* < 0.001.

### 3.6. Mental Health

A total of 98 students responded to the open-ended question about whether they have a mental health condition. Twenty-eight students (28.6%) reported having a diagnosed mental health condition at the time of the survey. Of these, eleven students (39.3%) reported an anxiety disorder, six students (21.4%) reported a depressive disorder, and eleven students (39.3%) reported a mix of mental health conditions, including both anxiety and depression, eating disorders, post-traumatic stress disorder (PTSD), and attention deficit hyperactivity disorder (ADHD). A total of 59 students (60.2%) reported not having a mental health condition and eight students (8.2%) were not sure about their mental health status. Of the 28 students with diagnosed conditions, 18 students (64.2%) reported they were able to access treatment either as usual or via alternative means (telehealth), whilst six students (21.4%) reported not being able to access care/medication due to the pandemic. According to their student status, there were 19 domestic (30.2%) and 9 international students (25.7%) who reported having a mental health condition.

### 3.7. Sleep

A total of 23 students (23.47%) reported very poor or poor sleep quality in the past two weeks, 39 students (39.8%) reported fair sleep quality and 36 students (36.7%) rated their sleep quality as good or very good. In terms of amount of sleep in the past two weeks, 12 students (12.2%) reported never or rarely getting enough sleep, 34 students (34.7%) sometimes, and 52 students (53.0%) most of the time or always getting enough. A total of 33 students (32.2%) had a disturbed sleep routine, with 38 students (38.8%) reporting sometimes adhering to a regular sleep routine and 28 students (28.5%) reporting adhering to their regular sleep routine.

### 3.8. Employment and Support Payments

A total of 51 students (48.6%) reported that they had worked in the past two weeks. Twenty-seven students (20.5%) reported they had lost their job as a result of the COVID-19 pandemic (fifteen international students and twelve domestic students). Conversely, fifteen students (14.3%) stated they had started a new job (six international students and nine domestic students). A total of 29 domestic students stated that they were receiving COVID-related support from the government (job-keeper/job-seeker) and one international student reported accessing a portion of their retirement savings (superannuation) via an available scheme [38]. International students were eligible for the COVID-19 early release of superannuation but not eligible for COVID-19 government support payments. A few international students (*n* = 7) received support from the University, such as emergency food vouchers and COVID-19 financial assistance, either as student loans which were repayable, or as grants up to $2000.

### 3.9. Support for International Students

In response to a final open-ended question about what government support should be provided to international students, 24 participants recorded a response. There was a range of answers including basic necessities, food, housing and amenities, emotional support and wellbeing support, reduced university fees, and free healthcare during the pandemic. The most frequent (*n* = 21) response was for financial aid to be provided.

## 4. Discussion

In this study, we aimed to explore the experiences of university students at an Australian university during the early stages of the COVID-19 crisis with respect to psychological distress and food insecurity. Our results showed a high level of both food insecurity (41.9%) and psychological distress (52.2%) among university students. We also found notable differences in food insecurity status and psychological distress levels between domestic students and international students, with international students much more likely to have a higher level of food insecurity and significantly higher levels of psychological distress than domestic students. Furthermore, the study results also demonstrated that food insecurity was significantly associated with psychological distress among university students.

Several studies have found that the proportion of university students reporting a higher food insecurity status category increased due to the COVID-19 pandemic [21,39]. In Australia, food insecurity among university students has been a persistent problem, even before the COVID-19 pandemic. A 2013 study of Queensland-based university students found one-quarter of students to be food-insecure [40], and in a 2019 study from the University of Newcastle, almost half of university students were found to be food-insecure [41]. Given concerns about the expected increase in food insecurity among university students during the pandemic, this study found a much higher level of food insecurity among international students (74.4%) compared to domestic students (22.7%), indicating that international students are more vulnerable than domestic students. The finding correlates with a cross-sectional study conducted at the University of Tasmania in March 2020 which found international students to have higher levels of food insecurity than domestic students (54% vs. 34%) [42].

A 2022 scoping review identified new food-insecure populations due to the COVID-19 pandemic in Australia, including international students [43]. The review suggested that international students became more vulnerable to food insecurity due to changes in employment, reduced financial support from family, and ineligibility for government welfare [43]. Similarly, a qualitative study in the US found the major cause of food insecurity to be income loss [20], where a greater proportion of international students (46.9%) lost their job during the COVID-19 pandemic compared with domestic students (21.8%), which may provide an explanation for the high rate of food insecurity in the current study. In response to the crisis, some Australian universities offered financial support, but this was very limited, especially for international students [44]. International students in Australia were also not eligible to receive government financial assistance and support in 2020 when the survey was conducted.

Another characteristic associated with changes in food security status among university students during the COVID-19 pandemic is that some domestic students were able to return to live in their family homes if necessary [19,21]. Two studies in the US suggested that students moving in with family improved their food security status during the COVID-19 pandemic [19,21]. Furthermore, at the time in the US, it was often not possible for international students to return to their home country due to border closures and a reduced number of international flights [45]. Similarly, the Australian Government closed the international border to all non-citizens and non-residents to limit the spread of the coronavirus on 20 March 2020 [4], and international students were given few alternatives but to return home or stay and support themselves [46]. In response to the crisis facing international students, local and state governments (Study NSW, Study Melbourne, and Study Adelaide), in partnership with local non-government organizations (NGOs) such as Foodbank, provided thousands of emergency food hampers which contained breakfast cereal, pasta, milk, canned fruit and vegetables [47,48,49]. Additionally, some universities provided emergency financial grants, including grocery e-vouchers [50].

Alarmingly, we found that over half of participants in this study had high or very high levels of psychological distress (52.2%). A study by Biddle et al., found that the levels of psychological distress increased between February 2017 and April 2020 for the Australian population, particularly for young adults aged 18 to 34 years [51]. Likewise, a scoping review of mental health in Australia before and during COVID also suggested that younger adults experienced greater declines in mental health than older adults during the COVID-19 pandemic [52]. A cross-national study in nine countries found the prevalence of high stress among university students was 61.3% in the total sample [53]. This study also noted that international students experienced higher stress than the general population [53], which is in line with our findings. In terms of student status, a previous study of university students in Australia showed that there was no difference in well-being between international and domestic students [10]. However, uncertainty about the future was significantly higher in international students than in domestic students [10]. In contrast, this study showed a significant association between student status and psychological distress. A recent systematic review indicated a significant increase in the prevalence of depression and anxiety during the COVID-19 pandemic among university students, with a higher prevalence in both mental health issues among university students in China [54]. The authors suggested that the differences in the prevalence of depression and anxiety are attributable to insufficient attention being paid to prevention of the pandemic at the initial stage of the outbreak [54]. A qualitative study at West Virginia University in the USA revealed that international students’ sentiment of homesickness and loneliness was further accentuated by the uncertainty due to travel restrictions [25].

In this study, food insecurity was significantly associated with psychological distress among university students. This finding is consistent with the previous cross-sectional study among undergraduate students in the Faculty of Health Sciences at the University of Ontario Institute of Technology, Canada [55]. A 2020 review suggested that a significant and positive association between food insecurity and psychological distress exists in certain population groups, including college students [56]. A growing body of research has suggested that food insecurity is a major social determinant of health and may negatively impact the mental wellbeing of university students [57,58,59]. Furthermore, poor mental health has been identified as a mediator of the association between food insecurity and academic achievement among university students [60].

In terms of learning and academic outcomes, although most students responded that they were able to manage planning their studies, nearly half of them perceived that they had failed to do well. It may be the case that students’ achievements, engagement, and perceptions of success decreased during the COVID-19 pandemic due to the challenges of online learning [61]. For example, a recent study in Cyprus reported that students had a significant reduction in engagement with the course and involvement in their desired subjects in online classes during the pandemic [62]. Further evidence suggests that academic disruption leads to loss of motivation for study, impacts daily life, and increases the dropout rate among students [26,63].

As a consequence of the COVID-19 pandemic and related disruptions, university students also have experienced changes in sleeping habits; however, the cross-sectional nature of our data precludes us from attributing these to the pandemic. A total of 22.5% of students reported having very poor or poor sleep quality, and almost 12% of students reported never or rarely getting enough sleep. A study during early lockdown in Jordan observed that the impact of the extended quarantine in university students negatively affected their sleeping habits [64]. Despite having distance-learning and asynchronous classes, most students had shifted to later bedtimes, reduced sleep duration at night and reduced sleep efficiency [65]. Interestingly, a US study that investigated sleep behaviour before and during stay-at-home orders in 139 university students found that overall, the students met the minimum seven hours per night of sleep, which is recommended by the Centers for Disease Control and Prevention [66].

### 4.1. Implications

Our findings show a high prevalence of food insecurity and psychological distress among university students during the COVID pandemic in 2020. Furthermore, our findings support the emerging evidence that levels of food insecurity and psychological distress remain higher among international students in Australian universities. Despite of these persistent issues, international students were often neglected because there was limited assistance available from Australian universities and the government during the COVID-19 pandemic. These raise questions about the levels of awareness of the issues within Australian universities and the government. Food banks and vouchers are the most widely adopted strategies by Australian universities to provide food relief for international students in times of crisis. However, a number of studies highlight that food hampers may not provide a healthful diet [67] and adequately meet the dietary intakes of university students [68]. Therefore, an effective and integrated long-term solution is greatly needed to address the food insecurity issues.

There is a need to conduct a regular university-wide monitoring of food insecurity and psychological distress among university students. Co-designing approaches can be adopted to design interventions to improve food security and psychological distress involving university students and university stakeholders. This provides an opportunity for universities to pilot potential initiatives and assess feasibility and acceptability. Shifting from a food relief reliance model to food social enterprises can be a promising initiative to increase availability and improve accessibility to fresh food, such as fruit and vegetables, on campus [31]. Furthermore, large-scale interventions can be implemented in the university sector with local and state government support. Another key strategy for the higher education sector is to establish an overarching food policy and food sustainability on campus. Encouraging a holistic well-being program, including mindfulness, emotion regulation, and a self-awareness approach can be effective in decreasing levels of psychological distress [69]. The program may not only involve professional counselors, but also innovative methods such as peer-to-peer support, involvement of pets, and apps/web-based support.

### 4.2. Strengths and Limitations

The strengths of our study include using validated tools to measure food insecurity and psychological distress. Moreover, it looked closely at the food insecurity status and psychological distress between domestic and international students during the early COVID pandemic in 2020. This study has several limitations. As stated earlier, the cross-sectional design prevented attributing the results directly to the COVID-19 pandemic, as there was no information collected on food security and psychological distress using these instruments before the pandemic in this population. Secondly, there is likely to be a degree of self-selection bias, as the decision to participate in the survey may reflect an inherent bias in the characteristics of the university students who agreed to participate. Thirdly, participants from health and medicine backgrounds were also over-represented in the sample, so the results may not be generalisable to all Australian university students. Another limitation is that the sample size was small. This may have been due to a reluctance to complete the survey due to the time required (approximately 20–30 min) or other competing demands. Lastly, the questions on mental health conditions in this survey was not able to differentiate between existing and new conditions. Therefore, it is not clear whether the COVID-19 pandemic has increased the incidence or exacerbation of mental health conditions, such as generalised anxiety and depression, or captured existing conditions.

## 5. Conclusions

This study provides further evidence that there was a high level of food insecurity and psychological distress among university students, particularly international students, during the early stages of the COVID-19 pandemic in Australia. This study also demonstrates a positive association between food insecurity and psychological distress in this student population, which emphasises a strong call for all stakeholders in the Australian education sectors to take action to address food insecurity issues and provide appropriate well-being support to university students. Future studies should use qualitative methods to further understand experiences during this turbulent period, which will likely assist government and educational institutions to design appropriate financial and psychological response packages during similar crises. This will no doubt be welcome among all university students, but particularly among vulnerable populations which include international students.

## Figures and Tables

**Table 1 ijerph-19-14101-t001:** Sociodemographic characteristics of the students (N = 105).

Characteristics	Domestic Students (*n* = 66) ^a^	International Students (*n* = 39) ^a^
Gender		
Male	9 (13.8%)	6 (15.4%)
Female	55 (84.6%)	33 (84.6%)
**Age group**		
<20	20 (31.3%)	3 (7.9%)
20–24	26 (40.6%)	7 (18.4%)
25–29	6 (15.6%)	18 (57.9%)
30+	8 (12.5%)	6 (15.8%)
**Citizenship**		
Australian citizen	61 (93.8%)	0 (0%)
Permanent resident	3 (4.6%)	0 (0%)
Visa holder	0 (0%)	35 (89.7%)
Other	1 (1.5%)	4 (10.3%)
**Language spoken at home**		
English	55 (84.6%)	9 (23.1%)
Hindi/Indian languages	0 (0%)	2 (5.1%)
Filipino/Tagalog	0 (0%)	6 (15.4%)
Cantonese/Mandarin	3 (4.6%)	0 (0%)
Arabic	0 (0%)	3 (7.7%)
Other	7 (10.8%)	19 (48.7%)
**Education**		
Postgraduate	26 (39.4%)	34 (87.2%)
Undergraduate	39 (59.1%)	5 (12.8%)
**Stage of Course (Year)**		
1	19 (30.2%)	17 (44.7%)
2	27 (42.9%)	17 (44.7%)
3	14 (22.2%)	3 (7.9%)
4+	3 (4.8%)	1 (2.6%)
**Enrolment status**		
Full-time	59 (89.4%)	37 (94.9%)
Part-time	7 (10.6%)	2 (5.1%)
**Faculty**		
Science and Engineering	9 (13.6%)	5 (12.8%)
Medicine, Health and Human Sciences	49 (74.2%)	31 (79.5%)
Arts	7 (10.6%)	1 (2.6%)
Macquarie Business School	1 (1.5%)	2 (5.1%)
**Employment status**		
Unemployed	24 (36.4%)	20 (51.3%)
Casual	27 (40.9%)	13 (33.3%)
Part-time	9 (13.6%)	5 (12.8%)
Full-time	5 (7.6%)	0 (0%)

Note: Data collected from 16 July 2020 to 20 September 2020; ^a^ Numbers may vary due to missing values.

**Table 2 ijerph-19-14101-t002:** Impacts on study in the last semester (N = 105).

Agreement with Following Statements	Domestic Students (*n* = 66)	International Students (*n* = 39)	*p* Value
I managed to plan my studies so that they were done on time	46 (73.0%)	29 (76.3%)	*p* = 0.713
I needed to invest more time and/or effort than usual to do my coursework	48 (78.7%)	31 (86.1%)	*p* = 0.364
I sometimes did nothing while I should have been doing my coursework	53 (82.8%)	25 (69.4%)	*p* = 0.121
I did really well in my studies	38 (65.5%)	16 (43.2%)	*p* = 0.033 *
I had the resources I needed to do my coursework well (technology, support)	55 (87.3%)	20 (55.6%)	*p* < 0.001 *
I had a clear understanding of what I was expected to do	43 (68.3%)	21 (56.8%)	*p* = 0.248
I had difficulty keeping up	38 (62.3%)	24 (61.5%)	*p* = 0.939

* Significant association.

**Table 3 ijerph-19-14101-t003:** Twelve-month prevalence of food insecurity indicators (N = 105).

Food Security Scale Items		Past 12 Months	
		n	%
The food that I bought just didn’t last, and I didn’t have money to get more	Often	6	5.7
	Sometimes	23	21.9
	Never	71	67.6
	Prefer not to answer	5	4.8
I couldn’t afford to eat balanced meals	Often	15	14.3
	Sometimes	21	20
	Never	66	62.9
	Prefer not to answer	3	2.9
I skipped meals because there wasn’t enough money for food	Yes	24	22.9
	No	75	71.4
	Prefer not to answer	6	5.7
If yes (skip meals), how often did this happen? ^a^	Almost every month	8	7.6
	Some months but not every month	9	8.6
	Only 1 or 2 months	4	3.8
	Prefer not to answer	3	2.9
Did you eat less than you felt you should because there wasn’t enough money for food?	Yes	24	22.9
	No	77	73.3
	Prefer not to answer	4	3.8
Were you ever hungry but didn’t eat because there wasn’t enough money for food?	Yes	15	14.3
	No	85	81
	Prefer not to answer	5	4.8

^a^ Numbers may vary due to conditioning question.

**Table 4 ijerph-19-14101-t004:** Levels of food security (N = 105).

Level of Food Security	Domestic Students N (%)	International Students N (%)	Total N (%)
Food secure	51 (77.3%)	10 (25.6%).	61 (58.1%)
Low food security	13 (19.7%)	23 (59.0%)	36 (34.3%)
Very low food security	2 (3.0%)	6 (15.4%)	8 (7.6%)

**Table 5 ijerph-19-14101-t005:** Level of psychological distress (N = 92).

Level of Psychological Distress	Domestic Students (*n* = 62)	International Students (*n* = 30)	Total N (%)
Little/no distress	14 (22.6%)	3 (10.0%)	17 (18.5%)
Moderate	20 (32.3%)	7 (23.3%)	27 (29.3%)
High	16 (25.8%)	7 (23.3%)	23 (25.0%)
Very high	12 (19.4%)	13 (43.3%)	25 (27.2%)

## Data Availability

Data sharing is not applicable.

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
