# Peer review of "Higher Prevalence of Food Insecurity and Psychological Distress among International University Students during the COVID-19 Pandemic: An Australian Perspective"

_ijerph, 2022, doi:10.3390/ijerph192114101_

Round 1

Reviewer 1 Report

Dear authors,

The topic of your paper Higher prevalence of food insecurity and psychological distress among international University students during the COVID-19 3 pandemic: an Australian perspective is quite interesting, however, there are some significant issues to consider:

Introduction. Lines 45-49. It is not now clear whether students were affected by what?

2.1. Design and participants. The number of participants, distribution between genders, and age range would make sense to add to this paragraph.

2.2. Data collection and 2.3. Data analysis and coding. I suggest the information regarding measures presented in these parts to combine in separate subsection Measures and present there the instruments and their respective codings.

Results. The first paragraph in the Results could be moved to Design and participants section.

The sample size is too small to make a sound conclusion on the research question. Also, the level of generalizability of the results could be strengthened.  

Author Response

Manuscript ID: ijerph-1913271

Title: Higher prevalence of food insecurity and psychological distress among international University students during the COVID-19 pandemic: an Australian perspective

Dear Editor,

Many thanks for sending through the reviewers’ comments on our manuscript (ID: ijerph-1913271) titled Higher prevalence of food insecurity and psychological distress among international University students during the COVID-19 pandemic: an Australian perspective. We thank the reviewer for their suggestions and feedback. We have made the changes suggested and included a point-by-point response below.

Response to Reviewer 1

We thank the reviewer for his/her thoughtful comments. We have addressed the comments below:

Reviewer comment The topic of your paper Higher prevalence of food insecurity and psychological distress among international University students during the COVID-19 pandemic: an Australian perspective is quite interesting, however, there are some significant issues to consider.

Authors response: Thank you.

Reviewer comment Introduction. Lines 45-49. It is not now clear whether students were affected by what?

Authors response: We have removed the sentence that students were affected and we go on to explain how they were impacted in terms of closure of educational institutions. “Learning and teaching delivery was shifted from face-to-face to online mode. There is evidence that these changes resulted in detrimental effects on learning, health and wellbeing. Some studies report inadequate achievement of learning outcomes for some students due to the shift from face-to-face to online modes [10-12].” (Line 46-50) This gives a better indication of how the COVID-19 lockdown affected students.  

Reviewer comment Design and participants. The number of participants, distribution between genders, and age range would make sense to add to this paragraph.

Authors response: We have added distribution between genders and age range. “Student ages ranged from 18 to 59 years, with 84.6% female (88/105).” (Line 102-103). The number of participants is written in Line 102.

Reviewer comment Data collection and 2.3. Data analysis and coding. I suggest the information regarding measures presented in these parts to combine in separate subsection Measures and present there the instruments and their respective codings.

Authors response: We agree with reviewer’s suggestion to present the information in separate subsections. Data analysis and coding has three subsections, namely demographic and impacts of COVID-19, food security, and psychological distress (Line 122-143)

Reviewer comment Results. The first paragraph in the Results could be moved to Design and participants section.

Authors response: We agree and have moved this paragraph to design and participants section (Line 98-102)

Reviewer comment The sample size is too small to make a sound conclusion on the research question. Also, the level of generalizability of the results could be strengthened.  

Authors response: In the discussion (strengths and limitations sub-section), we have outlined that the sample size is small and that further research is needed in this area to determine the links between food insecurity and psychological distress. We also acknowledge that the results may not be generalisable to all Australian university students (Line 420-423).

Reviewer 2 Report

I congratulate the authors on making a significant and potentially important contribution towards our understanding of the plight of international students during the Covid-19 pandemic. 

Overall, I found the study interesting and the topic worthy of investigation. I believe the manuscript will be improved by implementing the following points, I think firstly the abstract should be more succinct. It is writing a very conflated way at the moment, and should be more precise in its wording.

 The literature review could be improved, it makes a few key points about the disruptions of Covid, and then a few points about how international students were affected globally. However the statement you make whereby you write: 'In the Australian context, little is known about how the COVID-19 pandemic has impacted university students with regard to online learning, social connection, employment, and balancing financial responsibilities associated with housing and food', is incorrect. There is a range of published work specific to the effects Covid has had on international students mental health/social practices in Australia. And this should be included. See for example: Humphrey, A., & Forbes-Mewett, H. (2021). Social Value Systems and the Mental Health of International Students During the COVID-19 Pandemic . Journal of International Students11(S2), 58–76. https://doi.org/10.32674/jis.v11iS2.3577

The work of Catherine Gomes would also be relevant here. 

The analysis methods are sound. And results are well described and well detailed.

Overall, what was present in the discussion was well-written. However there were a number of places that the authors could have expanded more. For instances, I would have liked to see more information about what implications these findings have for Australian Universities in the future. 

Author Response

Manuscript ID: ijerph-1913271

Title: Higher prevalence of food insecurity and psychological distress among international University students during the COVID-19 pandemic: an Australian perspective

Dear Editor,

Many thanks for sending through the reviewers’ comments on our manuscript (ID: ijerph-1913271) titled Higher prevalence of food insecurity and psychological distress among international University students during the COVID-19 pandemic: an Australian perspective. We thank the reviewer for their suggestions and feedback. We have made the changes suggested and included a point-by-point response below.

Response to Reviewer 2

We thank the reviewer for his/her thoughtful comments. We have addressed the comments below:

Reviewer comment I congratulate the authors on making a significant and potentially important contribution towards our understanding of the plight of international students during the Covid-19 pandemic. 

Authors response: Thank you.

Reviewer comment Overall, I found the study interesting and the topic worthy of investigation. I believe the manuscript will be improved by implementing the following points, I think firstly the abstract should be more succinct. It is writing a very conflated way at the moment, and should be more precise in its wording.

Authors response: We have edited the abstract substantially and believe that it reads better now (Line 10-12; 22-27)

Reviewer comment The literature review could be improved, it makes a few key points about the disruptions of Covid, and then a few points about how international students were affected globally. However the statement you make whereby you write: 'In the Australian context, little is known about how the COVID-19 pandemic has impacted university students with regard to online learning, social connection, employment, and balancing financial responsibilities associated with housing and food', is incorrect. There is a range of published work specific to the effects Covid has had on international students mental health/social practices in Australia. And this should be included. See for example: Humphrey, A., & Forbes-Mewett, H. (2021). Social Value Systems and the Mental Health of International Students During the COVID-19 Pandemic . Journal of International Students11(S2), 58–76. https://doi.org/10.32674/jis.v11iS2.3577. The work of Catherine Gomes would also be relevant here. 

Authors response: We agree that there is now more evidence about how COVID-19 has impacted Australian students. We have edited the statement as “In the Australian context, there is emerging evidence about the flow on effects of disruptions on the mental health of university students [10,28-30]. In regard to food security, there is also recent evidence that the COVID-19 pandemic has impacted domestic and international students [31,32]. A few studies elsewhere show the disproportionately greater impact on the international student cohort [19,20]. A few studies elsewhere show the disproportionately greater impact on the international student cohort [19,20].” (Line 74-78). We also have added more relevant references. (Line 78)

Thank you for providing relevant references.

Reviewer comment The analysis methods are sound. And results are well described and well detailed.

Authors response: Thank you.

Reviewer comment Overall, what was present in the discussion was well-written. However there were a number of places that the authors could have expanded more. For instances, I would have liked to see more information about what implications these findings have for Australian Universities in the future. 

Authors response: We have added implications sub-section in discussion section (Line 381-408). In conclusions section, “This study provides further evidence that there was a high level of food insecurity and psychological distress among university students, particularly international students, during the early stages of the COVID-19 pandemic in Australia. This study also demonstrates a positive association between food insecurity and psychological distress in this student population, which emphasises a strong call for all stakeholders in the Australian education sectors to take actions to address food insecurity issues and provide appropriate well-being support to university students.” (Line 431-437)
